# Comparison of Vitamin D Levels, Bone Metabolic Marker Levels, and Bone Mineral Density among Patients with Thyroid Disease: A Cross-Sectional Study

**DOI:** 10.3390/diagnostics10121075

**Published:** 2020-12-11

**Authors:** Masliza Hanuni Mat Ali, Tuan Salwani Tuan Ismail, Wan Norlina Wan Azman, Najib Majdi Yaacob, Norhayati Yahaya, Nani Draman, Wan Mohd Izani Wan Mohamed, Mohd Shafie Abdullah, Hanim Afzan Ibrahim, Wan Nor Fazila Hafizan Wan Nik, Mafauzy Mohamed

**Affiliations:** 1Endocrine Unit, Department of Medicine, School of Medical Sciences, Universiti Sains Malaysia Health Campus, Kubang Kerian 16150, Kelantan, Malaysia; hmasliza@gmail.com (M.H.M.A.); izani@usm.my (W.M.I.W.M.); mafauzy@usm.my (M.M.); 2Department of Chemical Pathology, School of Medical Sciences, Universiti Sains Malaysia Health Campus, Kubang Kerian 16150, Kelantan, Malaysia; fazilahafizan@yahoo.com; 3Hospital Universiti Sains Malaysia, Kubang Kerian 16150, Kelantan, Malaysia; najibmy@usm.my (N.M.Y.); drnani@usm.my (N.D.); drshafie@usm.my (M.S.A.); afzankk@usm.my (H.A.I.); 4Department of Biostatistic, School of Medical Sciences, Universiti Sains Malaysia Health Campus, Kubang Kerian 16150, Kelantan, Malaysia; 5Endocrine Unit, Department of Medicine, Hospital Raja Perempuan Zainab 2, Kota Bharu 15200, Kelantan, Malaysia; jaiyati@yahoo.com; 6Department of Family Medicine, School of Medical Sciences, Universiti Sains Malaysia Health Campus, Kubang Kerian 16150, Kelantan, Malaysia; 7Department of Diagnostic and Radiology, School of Medical Sciences, Universiti Sains Malaysia Health Campus, Kubang Kerian 16150, Kelantan, Malaysia; 8School of Dental Sciences, Universiti Sains Malaysia, Kubang Kerian 16150, Kelantan, Malaysia

**Keywords:** hyperthyroid, hypothyroid, euthyroid, vitamin d, bone turn over markers, bone mineral density

## Abstract

Thyroid hormones have a catabolic effect on bone homeostasis. Hence, this study aimed to evaluate serum vitamin D, calcium, and phosphate and bone marker levels and bone mineral density (BMD) among patients with different thyroid diseases. This cross-sectional study included patients with underlying thyroid diseases (*n* = 64, hyperthyroid; *n* = 53 euthyroid; *n* = 18, hypothyroid) and healthy controls (*n* = 64). BMD was assessed using z-score and left hip and lumbar bone density (g/cm^2^). The results showed that the mean serum vitamin D Levels of all groups was low (<50 nmol/L). Thyroid patients had higher serum vitamin D levels than healthy controls. All groups had normal serum calcium and phosphate levels. The carboxy terminal collagen crosslink and procollagen type I N-terminal propeptide levels were high in hyperthyroid patients and low in hypothyroid patients. The z-score for hip and spine did not significantly differ between thyroid patients and control groups. The hip bone density was remarkably low in the hyperthyroid group. In conclusion, this study showed no correlation between serum 25(OH)D levels and thyroid diseases. The bone markers showed a difference between thyroid groups with no significant difference in BMD.

## 1. Introduction

Based on the incidence of thyroid disorders and thyroid autoantibodies, the prevalence rates of overt and subclinical hypothyroidism were 2.5 and 14.7 per 1000 population, respectively, in individuals living in the coastal communities of Malaysia. Meanwhile, those of overt and subclinical hyperthyroidism were 4.9 and 29.4 per 1000 population, respectively [1]. Hormones play an essential role in maintaining homeostasis and regulating bone density. Hyperthyroidism and hypothyroidism are correlated with low BMD and higher risk of fracture [2]. In the study of Maccagnano G et al. (2016), the incidence rates of fragility fractures were 4.5% and 3.7% in individuals with hyperthyroidism and hypothyroidism, respectively [3].

Malaysia’s clinical practice guideline on the management of osteoporosis, which was published in 2010, recommendeds the use of hyperthyroidism as an indicator of low BMD. In India, hyperthyroid patients with concomitant vitamin D deficiency had a lower BMD than those with adequate vitamin D levels [4]. Low calcium and vitamin D levels in this group of patients may increase the risk of osteoporosis. No standard consensus had shown that calcium or vitamin D supplementation is effective for the treatment of thyroid disorders. Whether vitamin D insufficiency predisposes to thyroid disease and is associated with bone loss is a matter of debate. Calcium and vitamin D deficiencies are a global health problem. Recent studies have shown that vitamin D deficiency is common even in tropical countries, such as Vietnam, Malaysia, and Indonesia [5,6,7]. In Malaysia, studies have only assessed the vitamin D status of school children [7], Malay employees of an academic institution [6], women of child-bearing age [8], and postmenopausal women [9]. Therefore, this study aimed to evaluate and compare serum vitamin D, calcium, and phosphate and bone turnover marker levels and bone mineral density among patients with different thyroid disorders.

## 2. Materials and Methods

This cross-sectional study was conducted at the Endocrine Clinic, Hospital Universiti Sains Malaysia (HUSM) and Hospital Raja Perempuan Zainab (2), Kota Bharu, Kelantan, Malaysia. In total, 199 participants (*n* = 135, those with thyroid diseases and *n* = 64, healthy controls) aged between 20 and 40 years were enrolled in this research from June 2017 to July 2018. Written informed consent was obtained from all patients. The study was conducted based on the guidelines for good clinical practice, and the Human Research Ethics Committee of USM (USM/JEPeM/17010025) and the Malaysia Registry Ethical Committee (NMRR-16-2477-33191 [IIR]) approved this study on Jun and July 2017 respectively. 

The inclusion criteria were as follows: patients with hyperthyroidism, biochemically hyperthyroid state for at least 6 months with or without medications; those with hypothyroidism, biochemically hypothyroid state for at least 6 months with or without medications; those with euthyroid, biochemically euthyroid state; TSH and free T_4_ (fT_4_) concentrations within the adult’s reference interval (0.270–4.20 mIU/L and 12–22 pmol/L) for at least 6 months with or without medications. Participants with subclinical diseases were also included in this study. Patients in the control group did not present with any thyroid diseases. The exclusion criteria included individuals with chronic diseases (liver and renal impairment and diabetes); post-menopausal, lactating, and pregnant women; those on medication (OCP, steroids, calcium and vitamin D supplements, and antiepileptic drugs) affecting bone metabolism; those with chronic gastrointestinal disease, thyroid malignancy, and history of fracture within a year from the study period. The healthy controls were recruited among volunteers, consist of hospital staff and people living nearby to the hospital within a similar recruitment duration with thyroid disease patients. We advertised for volunteers through media social and memos.

The sample size for each objective was calculated using two population mean formulae [10]. We calculated the sample size based on the method used in previous studies. The calculation for serum 25-hydroxyvitamin D (25(OH)D) levels had the largest sample size.

The records of all patients with thyroid diseases had been assessed a day before follow-up. The most recent thyroid function test result (within 1 month before follow-up) was reviewed. The participants who fulfilled the criteria were contacted, and objectives as well as the rationale of the study was provided. The participants were instructed to fast overnight for 10–12 h prior to blood sampling to minimize variations in bone marker levels. On the appointment day, detailed information regarding the study was again provided to the participants, and blood samples were collected from patients who provided consent. The samples for the assessment of 25(OH)D, calcium, phosphate, procollagen type I N-terminal propeptide (P1NP), and carboxy terminal collagen crosslink (CTX) levels were sent immediately to the laboratory. After centrifugation, the serum was stored at −70 °C before analysis.

Serum calcium and phosphorus levels were assessed via a spectrophotometric analysis using the ARCHITECT C800 analyzer (Abbott Diagnostics). The corrected calcium level was calculated using the following formula: corrected calcium (mmol/L) = measured total calcium level (mmol/L) + 0.02 (40–serum albumin level (g/L)). The serum 25(OH)D level was evaluated using the paramagnetic particle, chemiluminescent immunoassay competitive principle with the Access 2 Immunoassay System analyzer (Beckman Coulter). This method is traceable to a Joint Committee for Traceability in Laboratory Medicine using isotope dilution mass spectrometry reference method procedure (RMP) developed at Ghent University [11,12]. This RMP is further traceable to the NIST SRM 2972. According to the World Health Organization and the Institute of Medicine (IOM), the optimum vitamin D level threshold was 50 nmol/L [13,14]. The participants were classified based on the serum 25(OH)D levels, which were as follows: vitamin D sufficiency (≥50 nmol/L), insufficiency (<50 nmol/L), and deficiency (<30 nmol/L). The cut off 30 nmol/L used to define vitamin D deficiency because it is associated with rickets and osteomalacia and requires a high dose of vitamin D supplementation [14]. 

The total P1NP coefficients of variance (CVs) were 1.2–3.0% and 1.7–4.1% for repeatability and intermediate precision, respectively. Meanwhile, the CTX’s CVs were 2.1–3.5% and 2.8–8.4% for repeatability and intermediate precision, respectively. The total P1NP, CTX, TSH, and FT4 levels were assessed using the automated Cobas e601 immunoassay analyzer. A sandwich assay was used to measure the total P1NP and CTX levels. The TSH and FT4 values were evaluated with the electro-chemiluminescent assay based on the sandwich and competitive principle.

The left hip and lumbar spine (L1–L4 vertebrae) BMD was assessed using dual-energy X-ray absorptiometry (DEXA) with the Bone Densitometer Hologic Discovery with a fan beam technology (Waltham, MA, USA). Absolute BMD was the measured parameter using the bone mineral content in grams over a two-dimensional projected area of the bone in cm^2^. Thus, the BMD unit was g/cm^2^ [15].

The total z-scores of the left hip and lumbar spine were considered as the representative values. The z-score was used to represent a bone density in average individuals of the same age and sex (using Bone Densitometer Hologic Discovery QDR). According to the International Society for Clinical Densitometry 2015, a z-score of ≤−2.0 was below the expected range according to age, and a z-score of >−2.0 was within the expected range based on age.

Data were subjected to statistical analysis using the Statistical Package for the Social Sciences software version 24. For numerical variables, the results were presented as mean and standard deviation (SD) for data with normal distribution or as median and interquartile range (IQR) for skewed data. For categorical variables, the results were presented as frequency (n) and percentage (%). According to the different groups, the baseline characteristics of the participants were compared using one-way analysis of variance (ANOVA) for numerical variables and chi-square test or Fisher’s exact test for categorical variables. The difference in each variable between groups was compared using one-way ANOVA, followed by post hoc multiple pairwise comparisons using the Bonferroni procedure if the assumption of homogeneity for variance was fulfilled or the Games–Howell procedure if the assumption of homogeneity for variance was violated. Age and sex were adjusted in the analysis using two-way analysis of covariance (ANCOVA) or factorial ANCOVA, followed by multiple pairwise comparisons using the Bonferroni procedure. A *p* value *<* 0.05 was considered statistically significant.

## 3. Results

### 3.1. Baseline Characteristics of the Participants According to Group

Patients with thyroid diseases and healthy controls (*n* = 199) participated in this study. This research included euthyroid, hyperthyroid, and hypothyroid patients and healthy controls with a mean age (SD) of 31.77 (5.27), 29.69 (5.95), 28.89 (5.9), and 28.55 (6.17), respectively. The result was statistically significant. Most participants were Malay [*n* = 61 (93.5%), hyperthyroid; *n* = 17 (94.4%), hypothyroid; *n* = 49 (92.5%), euthyroid; *n* = 60 (93.7%), healthy controls), and only few were Chinese (*n* = 3 (4.7%), hyperthyroid; *n* = 1 (5.6%), hypothyroid; *n* = 4 (7.5%), euthyroid; *n* = 4 (6.3%), healthy controls]. The euthyroid group had a higher body mass index (BMI) [mean (SD): 25.79 (5.61) kg/m^2^] than the hypothyroid [24.43 (6.08) kg/m^2^] and hyperthyroid [23.69 (3.92) kg/m^2^] groups. The BMI of the control group was 24.44 (4.60) kg/m^2^.

The duration of illness was longer in the hypothyroid group [median (IQR): 49.00 (70.00) months] than in euthyroid [42.00 (63.00) months] and hyperthyroid group [25.50 (42.00) months]. The median (IQR) for TSH levels of the hyperthyroid, hypothyroid, euthyroid, and control groups were 0.005 (0.001), 31.97 (86.01), 1.60 (2.50) and 1.57 (1.08) mIU/L, respectively. Meanwhile, the median (IQR) for fT4 levels were 28.52 (14.92), 8.64 (12.12), 16.91 (4.13), and 15.82 (2.61) pmol/L in the hyperthyroid, hypothyroid, euthyroid, and control groups, respectively (Table 1). 

Graves’ disease was the most common diagnosis (*n* = 51 [80%], hyperthyroid; *n* = 11 [61%], hypothyroid; *n* = 38 [72%], euthyroid), followed by toxic goiter (*n* = 13 [20%], hyperthyroid; *n* = 1 [6%], hypothyroid; *n* = 10 [19%], euthyroid) and thyroiditis (*n* = 6 [33%], hypothyroid; *n* = 4 [8%], euthyroid). In this study, only one patient had congenital hypothyroidism.

In total, 53 (83%) patients in the hyperthyroid group were on anti-thyroid medications. Of them, 11 (17%) received radioactive iodine therapy and anti-thyroid medications. Moreover, 10 (56%) hypothyroid patients were treated with radioactive iodine and required thyroid hormone replacement. Six (33%) patients required thyroid hormone replacement therapy alone. Among them, two (11%) underwent thyroidectomy and required thyroxine replacement therapy after surgery. 

In the euthyroid group, 29 (55%) patients achieved euthyroid state with anti-thyroid therapy, and 12 (23%) received radioactive iodine therapy. The patients required thyroxine replacement after radioiodine therapy. Five (9%) patients still required anti-thyroid medications after radioactive iodine therapy, and only one patient did not require any treatment after radioactive iodine therapy. Four (7%) patients did not undergo surgery but were on thyroxine replacement therapy or radioactive iodine before replacement therapy (Table 2). 

### 3.2. Assessment of Serum 25(OH)D Levels and Comparison of Serum 25(OH)D, Calcium, and Phosphate Levels among the Hyperthyroid, Hypothyroid, Euthyroid, and Control Groups 

#### 3.2.1. Serum 25(OH)D Levels

The serum 25(OH)D level was higher in men than in women in each group (Table 3). More than half of the participants (*n* = 145 [73%]) had vitamin D insufficiency (<50 nmol/L). The euthyroid group had the highest 25(OH) D levels (49.55 [18.57] nmol/L), followed by the hypothyroid group (45.74 [15.17] nmol/L) and the hyperthyroid group (43.6 [20.83] nmol/L). The results were statistically significant based on the one-way ANOVA (*p* = 0.006, Table 4). However, the control group had the lowest mean serum 25(OH) D level (37.38 [17.21] nmol/L). The post hoc pairwise comparison using the Games–Howell method revealed that the results significantly differed only in the euthyroid and control groups (mean difference [MD]: 12.17; 95% confidence interval [CI]: 2.94, 21.40; *p* = 0.003) (Table 5). Based on the analysis using ANCOVA adjusted for age and gender, the serum 25(OH)D levels remained significantly different (*p* = 0.030, Table 6). The post hoc pairwise comparison using ANCOVA adjusted for age and gender showed that the mean serum 25(OH)D levels differed significantly only between the euthyroid and control groups (MD: 13.94; 95% CI: 5.22, 22.67; *p* < 0.001) (Table 7).

#### 3.2.2. Serum Calcium Levels

The mean serum calcium level was within the 95% reference interval normal in all groups. The level was slightly lower in the thyroid disease group than in the control group. The hyperthyroid group had the lowest mean serum calcium level, followed by the euthyroid and hypothyroid groups. The results were statistically significant (*p* < 0.001). Based on the ANCOVA adjusted for age and gender, the serum calcium levels significantly differed (*p* < 0.001) (Table 6). The post hoc pairwise comparison using ANOVA for serum calcium revealed significant difference between hyperthyroid versus control groups (MD: –0.12; 95% CI: –0.18, –0.06, *p* < 0.001) and euthyroid versus control group (MD: –0.11, 95% CI: –0.18, –0.50, *p* < 0.001) (Table 5). The post hoc pairwise comparison using ANCOVA for serum calcium (adjusted for age and gender) revealed a significant difference only between hyperthyroid versus the control group (MD: –0.12; 95% CI: –0.19, –0.05, *p* < 0.001) (Table 7).

#### 3.2.3. Serum Phosphate Level

The mean serum phosphate level was within the 95% reference interval normal in all groups. Moreover, the control group had a higher mean serum phosphate level than the thyroid disease group (Table 4). Moreover, the hyperthyroid group had a higher serum phosphate level than the euthyroid and hypothyroid groups. The results were statistically significant based on the analysis using one-way ANOVA (*p* = 0.024). However, the post hoc pairwise comparison using the Games–Howell method revealed that the results did not significantly differ. Analysis using ANCOVA, adjusted for age and gender, revealed that the mean serum phosphate level did not significantly differ, and the results remained insignificant in the post hoc pairwise comparison using ANCOVA (Table 5, Table 6 and Table 7).

### 3.3. Comparison between Serum Bone Turnover Marker Levels among the Hyperthyroid, Hypothyroid, Euthyroid, and Control Groups

#### 3.3.1. Serum CTX) Levels

The serum CTX level was used as a bone resorptive marker and the serum P1NP level as a bone formation marker. The hyperthyroid group had the highest mean serum CTX levels, whereas the hypothyroid group had the lowest level. The results were statistically significant (*p* = 0.001, Table 8). The results were statistically significant based on the analysis using ANCOVA adjusted for age and gender (*p* = 0.002, Table 9). The post hoc pairwise comparisons using the Games–Howell method for ANOVA revealed a significant difference between the hyperthyroid and hypothyroid groups and control group (*p* < 0.001, Table 10). The post hoc pairwise comparison using ANCOVA adjusted for age and gender revealed that the difference in results remained significant between the hyperthyroid and hypothyroid groups (MD: 0.31, 95% CI; 0.07, 0.54; *p* = 0.028). However, the result was not significant between the hypothyroid and control groups. The hyperthyroid and euthyroid groups significantly differed after adjusting for age and gender (MD: –0.19; 95% CI: –0.42, 0.05, *p* = 0.205; Table 11).

#### 3.3.2. Serum P1NP Level

The mean serum P1NP level was used as a marker of bone formation. One-way ANOVA revealed that the hyperthyroid group had a higher mean serum P1NP level than the euthyroid and control groups. Further, the hypothyroid group had the lowest mean serum P1NP level. The results were statistically significant (*p* < 0.001, Table 8). Based on the analysis using ANCOVA adjusted for age and gender, the results were significant (*p* < 0.001, Table 9). The post hoc pairwise comparison using the Games–Howell method for ANOVA revealed a significant difference between the hypothyroid and control groups (MD: –26.48 ng/mL, 95% CI: –43.16, –9.8, *p* < 0.001, and between the hyperthyroid and the other groups (*p* < 0.001, Table 10). According to the pairwise comparison using ANCOVA adjusted for age and gender, the results between the hyperthyroid and the other groups were significant. However, no significant difference was noted between the hypothyroid and control groups (Table 11).

### 3.4. Comparison of BMD among the Hyperthyroid, Hypothyroid, Euthyroid, and Control Groups 

#### Z-Score/Bone Density (g/cm^2^)

Of 199 patients, only 186 underwent BMD measurement. Moreover, 13 participants did not attend the BMD appointment. Bone density (g/cm^2^) and z-score (bone density compared according to similar age and sex) of the left hip and spine were analyzed. Based on the analysis of z-score for left hip and spine using one-way ANOVA, the thyroid group was found to have lower hip z-scores than the control group (0.39 [1.03]). However, the results were not statistically significant (*p* = 0.164, Table 12). The post hoc pairwise comparison using the Bonferroni method for ANOVA revealed no significant difference among the hyperthyroid, hypothyroid, euthyroid, and control groups (Table 13). Based on the analysis of covariates and post hoc pairwise comparison with ANCOVA adjusted for age and gender, the results were not statistically significant among the groups (Table 14 and Table 15).

ANOVA for bone density (g/cm^2^) of the left hip and spine revealed that the control and euthyroid groups had a higher bone density than the hypothyroid and hyperthyroid groups. The results for the left hip bone density was statistically significant (*p* = 0.002, Table 12). The analysis using ANCOVA adjusted for age and gender revealed that the difference in the left hip bone density remained statistically significant (*p* = 0.011, Table 14). Based on the post hoc pairwise comparison using ANOVA and ANCOVA, the left hip bone density between the hyperthyroid and control groups differed significantly (Table 13 and Table 15).

## 4. Discussion

Thyroid diseases have widespread systemic manifestations. That is, they affect the bone. However, thyrotoxicosis and hypothyroidism have a different impact on bone metabolism. Hyperthyroidism is an important cause of secondary osteoporosis and is associated with significant bone loss at the hip, forearm, and lumbar spine [16]. Thyroid-stimulating hormones (TSH) directly affect bone remodeling via the TSH receptor, which is found on osteoblasts and osteoclast precursor cells [17].

Vitamin D plays a pivotal role in the development and maintenance of a healthy skeleton. Moreover, it plays a critical function in calcium and phosphorus metabolism and ensures an adequate level of these minerals for bone mineralization [18]. The serum 25(OH)D concentration is an indicator of vitamin D status. It reflects vitamin D level from endogenous production and food or supplements [19], and it has a relatively long circulating half-life (15 days) than 1,25(OH)2D (15 h) [20].

In this study, the 25(OH)D levels range from 12.26 to 111.22 nmol/L. However, the mean serum 25(OH)D levels in patients with thyroid diseases and healthy controls were below the adequate level (<50 nmol/L). A systematic review of vitamin D status in populations worldwide conducted in 2013 revealed that 37.3% of the participants had a vitamin D level <50 nmol/L, and 6.7% had values <25 nmol/L [21]. The mean vitamin D levels among individuals living in countries in the South East Asia region, such as Indonesia and Malaysia are 38.7 and 67.6 nmol/L, respectively [22,23]. Even though both are tropical countries, similar to Malaysia, the result significantly differed.

Factors, such as dietary intake and daily duration of sun exposure, seasonal variation, physical activity, rural/urban area, skin pigmentation, and sun-protective behavior can influence serum vitamin D level. In Malaysia, Nurbazlin et al. (2013) reported that the 25(OH)D concentration in women living in a rural area was significantly higher than that in women living in urban areas [24]. Seasonal variations affect serum 25(OH)D concentrations in patients with Graves’ disease. This result is supported by a study showing that serum 25(OH)D concentrations were high during summer and low during winter [25]. Another study showed a slight decrease in the mean serum vitamin D concentration during winter compared with during summer [26]. Nevertheless, this study involved thyroid patients from the East Coast, Malaysia, throughout the year. In Malaysia, there are almost no seasonal variations in weather. However, the sample collection might include November to February, which had more heavy rainfall and might contribute to the lower level of vitamin D.

The serum 25(OH)D concentration was higher in men than in women. Yamashita et al. (2005) reported the occurrence of hypovitaminosis D in 40% of women and 18% of men with Graves’ disease in Japan [25]. Meanwhile, Mackay AM et al. (2013) showed that the serum 25(OH)D level was lower in hypothyroid women than in hypothyroid men [27]. In Kuala Lumpur, Malaysia, a study on adults revealed that approximately 41% of men and 87% of women had insufficient vitamin D levels (<50 nmol/L) [6]. However, there is no clear evidence on differences in terms of gender. However, a large study on obese adult women revealed that the possible pathophysiology is the deposition of vitamin D in the subcutaneous fat, which is released when there is a decreased cutaneous production [28]. Nonetheless, in our study, the mean BMI among the participant commonly ranges from 23.69 to 25.75 kg/m^2^. This finding could not explain this condition. One cause is that men are more exposed to sunlight due to their clothing. By contrast, women preferred clothing that covers the entire body regardless of weather conditions. Other reasons include less sun exposure, few outdoor activities, and limited body area exposure in women than in men. Although there was a higher number of men than women in this study, the results of the current study were contrasting with those of other studies. These discrepancies might be attributed to the differences in the patient selection process, varying background characteristics, and residents of patients with thyroid diseases. 

This study also found that the mean serum 25(OH)D concentration was higher in patients with thyroid disease than in healthy controls. This finding was not in accordance with those of other studies conducted in different countries. For example, Wang, Jiying et al. (2015) revealed that thyrotoxic patients had a significantly lower plasma 25(OH) vitamin D levels than controls [29]. Moreover, Dhanwal et al. (2010) examined the vitamin D level in premenopausal hyperthyroid patients. The results showed that the 25(OD)D levels of 26.6% of patients with hyperthyroidism was L < 25.0 nmol/L, and 73.4% had sufficient vitamin D levels, with a mean (SD) of 46.0 (13.5) nmol/L [4]. Similarly, Mackawy AM et al. (2013) reported that the serum 25(OH)D level was significantly lower in hypothyroid patients than in controls [27]. However, several studies did not show a correlation between low vitamin D level and autoimmune thyroid diseases or Hashimoto thyroiditis. Goswami et al. reported that there was no correlation between vitamin D deficiency and anti-TPO positivity in 642 students, teachers, and staff in India. However, only a poor reverse association was noted between serum 25(Oh)D levels and anti-TPO (r = −0.08; *p* = 0.04) [30].

This study also found that the serum 25(OH)D level in healthy control males was lower than male thyroid disease patients. The possibility might be attributed to the number of healthy control participants, which comprise of 75% of hospital staff who had less exposure to sunlight. While another 25% were among the non-hospital staff who live around the city and work in the office. However, we do not assess the amount of sunlight exposure among participants in this study. 

All groups had normal mean serum total calcium and phosphate levels. However, the mean calcium levels were higher in the control group than in the thyroid disease group, and the phosphate level was higher in the control group than in the thyroid disease group. However, the result was not significantly different. The results of previous studies were heterogeneous. In the study of Dhanwal et al. (2010) conducted in India, hyperthyroid patients only presented with slightly high serum calcium levels. However, the difference was not significant, and the serum phosphate level was normal [4]. Anuj Modi et al. (2018) showed that the serum calcium and phosphate levels were significantly higher in the hyperthyroid group than in the control group [31]. Moreover, Sridevi et al. (2016) reported that the hypothyroid group had lower calcium levels and higher serum phosphate levels than the control group [32]. 

Overt hyperthyroidism leads to the acceleration of bone turnover and loss of bone mineral density by 10–20% mainly in the cortical bone [33,34,35]. The cycle of bone remodeling is shortened by almost 50% (from 200 to 113 days), which disturbs the proportion of bone formation and bone resorption [2]. The phase of bone formation is reduced by 2/3, which results in the loss of <10% of mineralized bone in one cycle [36]. Hypothyroidism causes hypometabolism in general. Bone formation mechanisms decelerate in 40% of bone resorption processes [2].

The negative impact on bone equilibrium is initially expressed as an increase or decrease in bone turnover markers. Our findings showed that the bone turnover markers are higher in hyperthyroid patients and lower in hypothyroid patients. The euthyroid and control groups had a similar level of bone turnover markers. The results indicate that hyperthyroid patients had accelerated bone resorption and formation, whereas hypothyroid patients had reduced bone turnover. Restoration to euthyroid level after treatment for hyperthyroid and hypothyroid can improve bone turnover.

Few studies showed the persistent suppression of TSH levels, which results in a continuous increase in bone turnover even after the normalization of serum FT3 and FT4 levels [37,38]. As a higher bone turn over resulted in a continuous accelerated bone loss [39,40,41,42], TSH suppression might be a risk factor for secondary osteoporosis in patients with Graves’ disease. Therefore, to normalize bone metabolism with anti-thyroid drugs in patients with Graves’ disease, the serum TSH should be maintained within the reference range. The result of the current study on bone turnover was consistent with that of previous studies [37,39].

In the current study, bone densitometry showed that the z-score of the left hip and spine and bone density (g/cm^2^) of the spine were not significant. Only bone densities (g/cm^2^) of the left hip in hyperthyroid patients were significant. These findings differed from other studies. Rosario et al. (2008) showed a significant increase in the serum markers of bone formation and resorption in <65-year-old premenopausal women with subclinical hyperthyroid. In premenopausal patients, the bone mineral density in the femoral neck but not in the lumbar spine was lower. In contrast, other studies showed that postmenopausal patients had low BMD at both sites [43]. Tauchmanova et al. (2004) examined 60 patients with endogenous subclinical hyperthyroidism due to multinodular goiter. The study included 30 premenopausal and 30 early postmenopausal women. The results showed a significant decrease in femoral BMD both in pre- and postmenopausal women compared with controls. However, postmenopausal women had a higher femoral BMD [44]. Our study and the study of Tuchendler et al. (2013), which showed newly diagnosed hypothyroidism among premenopausal women (average age: 33.37 [10.83] years), did not influence bone density [45]. Similarly, Hanna et al. (1998) showed no difference in terms of bone mineral density in patients receiving replacement doses of thyroxine, irrespective of the etiology of hypothyroidism [46]. Demographic analyses of 4473 autoimmune hypothyroid participants have shown that the incidence of fractures increased during diagnosis [47]. However, the discovery of a possible pre-diagnosis correlation (i.e., before beginning T4 therapy) is in contrast to the role of excess thyroid hormones based on these results. However, there are still limited data on the association between the risk of fracture and T4 treatment, even at doses appropriate for TSH serum suppression. Moreover, precaution must be observed in high-risk groups, particularly postmenopausal women and, possibly, men [48]. However, reduced bone mineral density associated with hyperthyroidism may be restored, maintained, and, in some cases, improved while on long-term thyroxine replacement after radioiodine therapy [46].

The current study had several limitations. First, the characteristic of a cross-sectional study does not reflect results over time. Moreover, the sample size was small, particularly in the hypothyroid group. Most participants were Malay. Hence, this does not extrapolate the condition of other populations. The assessment of thyroid functions is based on a single assessment. Furthermore, in this study, we do not consider the variation in sunlight exposure according to season. Participants’ stratification by recruited season might produce a more valid outcome, especially in countries where the seasonal variation of sunlight is more prominent. Several other factors which may influence vitamin D level were not assessed in this study such as dietary intake and skin color. 

We recommend a prospective longitudinal study with a longer duration. Moreover, a more significant population should be included, and the study must be conducted at multiple centers. The assessment of average thyroid function better represents the stage of thyroid function within the last 6 months. Confounders correlated to vitamin D measurements, such as duration of daily sun exposure and dietary intake assessment, will provide more value to vitamin D status in patients with thyroid disease and control groups. 

## 5. Conclusions

In this study, serum 25(OH)D levels were not correlated with thyroid diseases. All groups had normal serum calcium and phosphate levels. The serum bone turnover marker levels were significantly high in the hyperthyroid group and low in the hypothyroid group. There was no difference in terms of z-score bone density among all groups. However, the bone density of the left hip was significantly low in the hyperthyroid group.

## Figures and Tables

**Table 1 diagnostics-10-01075-t001:** Baseline characteristic of patients with thyroid disease and control groups.

Variables	Hyperthyroid	Hypothyroid	Euthyroid	Control	Test Stat (df)	*p*-Value
**Frequency**	64 (32.0)	18 (9.0)	53 (27.0)	64 (32.0)		
**Age, years**	29.69 (5.95)	28.89 (5.90)	31.77 (5.27)	28.55 (6.17)	3.14 (3,195)	0.026 ^a^
**Gender**						
**Male**	16 (25.0)	5 (27.8)	11 (20.8)	27 (42.2)	7.53 (3)	0.057 ^b^
**Female**	48 (75.0)	13 (72.2)	42 (79.2)	37 (57.8)		
**Race**						
**Malay**	61 (95.3)	17 (94.4)	49 (92.5)	60 (93.7)	-	0.124 ^c^
**Non Malay**	3 (4.7)	1 (5.6)	4 (7.5)	4 (6.3)		
**BMI, kg/m^2^**	23.69 (3.92)	24.43 (6.08)	25.79 (5.61)	24.44 (4.60)	1.84 (3,195)	0.141 ^a^
**Duration of Thyroid Disease, Months**	25.50 (42.00)	49.00 (70.00)	42.00 (63.00)	-	7.16 (2)	0.028 ^d^
**TSH, mIU/L**	0.005 (0.001)	31.97 (86.01)	1.60 (2.50)	1.57 (1.08)	113.36 (3)	<0.001 ^d^
**fT4, pmol/L**	28.52 (14.92)	8.64 (12.12)	16.91 (4.13)	15.82 (2.61)	75.58 (3)	<0.001 ^d^
**Serum 25(OH)D**						
**Sufficient**	20 (31.3)	5 (27.8)	19 (35.8)	10 (15.6)	16.40 (6)	0.012 ^b^
**Insufficiency**	23 (35.9)	11(61.1)	27 (50.9)	30 (46.9)		
**Deficiency**	21 (32.8)	2 (11.1)	7 (13.3)	24 (37.5)		

Age and BMI were reported as mean (SD), duration of thyroid disease, TSH and fT4 were reported as median (IQR) for numerical variables and frequency, gender, race and serum 25(OH)D were reported as n (%). ^a^ One way ANOVA *p*-value, ^b^ Chi2 test *p*-value, ^c^ Fisher exact test *p*-value, ^d^ Kruskal–Wallis test *p*-value

**Table 2 diagnostics-10-01075-t002:** Diagnosis and Treatment According to Groups.

Variables	Hyperthyroid	Hypothyroid	Euthyroid
**Diagnosis**			
**Graves’ Disease**	51 (80)	11 (61)	38 (72)
**Toxic Goiter**	13 (20)	1 (6)	10 (19)
**Congenital Hypothyroid**	-	-	1 (2)
**Thyroiditis**	-	6 (33)	4 (7)
**Treatment**			
**ATD**	53 (83)	-	29 (55)
**RAI + ATD**	11 (17)	-	5 (9)
**Thyroidectomy + Thyroxine**	-	2 (11)	2 (4)
**RAI + Thyroxine**	-	10 (56)	12 (23)
**RAI**	-	-	1 (2)
**Thyroxine**	-	6 (33)	4 (7)

ATD: Antithyroid drug; RAI: radioactive iodine. Data described as n (%).

**Table 3 diagnostics-10-01075-t003:** Serum 25(OH)D Level In Between Groups, Comparing Male and Female.

Group		N	Mean (SD) (nmol/L)	Mean Difference (95% CI)	t-Stats (df)	*p*-Value
Hyperthyroid	Male	16	57.35 (28.02)	18.33 (2.88, 33.78)	2.49 (18.22)	0.023
	Female	48	39.02 (15.66)			
Hypothyroid	Male	5	63.48 (16.93)	24.57 (3.84, 45.3)	3.14 (4.53)	0.029
	Female	13	38.91 (6.98)			
Euthyroid	Male	11	63.6 (17.63)	17.73 (6, 29.46)	3.03 (51)	0.004
	Female	42	45.87 (17.16)			
Control	Male	27	44.82 (17.61)	12.85 (4.7, 21)	3.07 (50.39)	0.003
	Female	37	31.96 (14.93)			

Independent sample *t*-test.

**Table 4 diagnostics-10-01075-t004:** ANOVA analyses. Comparisons of the mean serum 25(OH)D, serum calcium and serum phosphate in between groups.

Variables (n)	Hyperthyroid (64)	Hypothyroid (18)	Euthyroid (53)	Control (64)	F-Stat (df)	*p* Value
25 (OH) D (nmol/L)	43.6 (20.83)	45.74 (15.17)	49.55 (18.57)	37.38 (17.21)	4.24 (3,195)	0.006
Serum Calcium (mmol/L)	2.20 (0.16)	2.21 (0.11)	2.21 (0.16)	2.32 (0.83)	10.64 (3,195)	<0.001
Serum Phosphate (mmol/L)	1.12 (0.24)	1.05 (0.18)	1.09 (0.25)	1.21 (0.24)	3.21 (3,195)	0.024

Data described as mean (SD).

**Table 5 diagnostics-10-01075-t005:** Post hoc pairwise comparison of ANOVA analysis for mean serum 25(OH)D, serum calcium, serum phosphate in between group, using Games–Howell method.

Group Comparison	Adjusted Mean Difference (95% CI)	*p*-Value
**Serum 25(OH)D**		
Hyperthyroid vs. Hypothyroid	−2.14 (−15.4, 11.12)	>0.95
Hyperthyroid vs. Euthyroid	−5.95 (−15.18, 3.28)	0.524
Hyperthyroid vs. Control	6.22 (−2.57, 15.00)	0.365
Hypothyroid vs. Euthyroid	−3.81 (−17.38, 9.75)	>0.95
Hypothyroid vs. Control	8.35 (−4.91, 21.62)	0.569
Euthyroid vs. Control	12.17 (2.94, 21.40)	**0.003**
**Serum Calcium**		
Hyperthyroid vs. Hypothyroid	−0.01 (−0.10, 0.08)	0.974
Hyperthyroid vs. Euthyroid	−0.01 (−0.09, 0.71)	0.994
Hyperthyroid vs. Control	−0.12 (−0.18, −0.06)	**<0.001**
Hypothyroid vs. Euthyroid	0.01 (−0.09, 0.10)	0.998
Hypothyroid vs. Control	−0.11 (−0.19, −0.03)	0.005
Euthyroid vs. Control	−0.11 (−0.18, −0.50)	**<0.001**
**Serum Phosphate**		
Hyperthyroid vs. Hypothyroid	0.08 (−0.09, 0.25)	>0.95
Hyperthyroid vs. Euthyroid	0.03 (−0.09, 0.15)	>0.95
Hyperthyroid vs. Control	−0.08 (−0.19, 0.03)	0.335
Hypothyroid vs. Euthyroid	−0.05 (−0.22, 0.13)	>0.95
Hypothyroid vs. Control	−0.16 (−0.33, 0.01)	0.082
Euthyroid vs. Control	−0.11 (−0.23, 0.01)	0.081

Data described as adjusted mean (95% CI).

**Table 6 diagnostics-10-01075-t006:** ANCOVA analysis adjusted for age and gender. Comparisons of the mean serum 25(OH)D, serum calcium and serum phosphate in between groups.

Variables	Hyperthyroid (64)	Hypothyroid (18)	Euthyroid (53)	Control (64)	F-Stat (df)	*p* Value
**25(OH) D** **(nmol/L)**	43.67 (39.16, 58.19)	46.32 (37.8, 54.58)	48.3 (43.26, 53.33)	38.19 (33.64, 42.74)	3.04 (3,194)	0.030
**Serum Calcium** **(mmol/L)**	2.21 (2.17, 2.24)	2.24 (2.16, 2.31)	2.22 (2.18, 2.27)	2.33 (2.29, 2.36)	8.322 (3,190)	<0.001
**Serum Phosphate** **(mmol/L)**	1.12 (1.06, 1.18)	1.04 (0.93, 1.15)	1.11 (1.04, 1.17)	1.2 (1.14, 1.25)	2.64 (3,194)	0.051

Data described as adjusted mean (95% CI).

**Table 7 diagnostics-10-01075-t007:** Post hoc pairwise comparison of ANCOVA analysis for mean serum 25(OH)D, serum calcium and serum phosphate adjusted for age and gender in between groups.

Group Comparison	Adjusted Mean Difference (95% CI)	*p*-Value
**Serum 25(OH)D**		
Hyperthyroid vs. Hypothyroid	−2.10 (−14.13, 9.92)	>0.95
Hyperthyroid vs. Euthyroid	−5.57 (−14.02, 2.89)	0.484
Hyperthyroid vs. Control	8.38 (0.28, 16.47)	0.038
Hypothyroid vs. Euthyroid	−3.47 (−15.87, 8.94)	>0.95
Hypothyroid vs. Control	10.48 (−1.59, 22.54)	0.130
Euthyroid vs. Control	13.94 (5.22, 22.67)	**<0.001**
**Serum Calcium**		
Hyperthyroid vs. Hypothyroid	−0.03 (−0.14, 0.08)	>0.95
Hyperthyroid vs. Euthyroid	−0.02 (−0.10, 0.06)	>0.95
Hyperthyroid vs. Control	−0.12 (−0.19, −0.05)	**<0.001**
Hypothyroid vs. Euthyroid	0.01 (−0.10, 0.13)	>0.95
Hypothyroid vs. Control	−0.09 (−0.20, 0.02)	0.136
Euthyroid vs. Control	−0.10 (−0.18, −0.02)	0.004
**Serum Phosphate**		
Hyperthyroid vs. Hypothyroid	0.08 (−0.09, 0.25)	>0.95
Hyperthyroid vs. Euthyroid	0.01 (−0.1, 0.13)	>0.95
Hyperthyroid vs. Control	−0.08 (−0.19, 0.04)	0.446
Hypothyroid vs. Euthyroid	−0.07 (−0.24, 0.11)	>0.95
Hypothyroid vs. Control	−0.16 (−0.33, 0.01)	0.077
Euthyroid vs. Control	−0.09 (−0.21, 0.03)	0.289

Data described as adjusted mean (95% CI).

**Table 8 diagnostics-10-01075-t008:** ANOVA analysis. Comparison between mean serum CTX and serum P1NP in between groups.

Variables *(n)*	Hyperthyroid (64)	Hypothyroid (18)	Euthyroid (53)	Control (64)	F-Stat (df)	*p* Value
**Serum CTX (ng/ml)**	0.48 (0.37)	0.19 (0.12)	0.29 (0.45)	0.39 (0.19)	5.53 (3,195)	**0.001**
**Serum P1NP (ng/ml)**	156.54 (114.37)	41.05 (19.3)	73.17 (114.39)	67.53 (34.61)	15.09 (3,195)	**<0.001**

Data described as mean (SD).

**Table 9 diagnostics-10-01075-t009:** ANCOVA analysis adjusted for age and gender. Comparison between mean serum CTX and serum P1NP in between groups.

Variables *N*	Hyperthyroid *(64)*	Hypothyroid *(18)*	Euthyroid *(53)*	Control *(64)*	F-Stat *(df)*	*p* Value
**Serum CTX** **(ng/ml)**	0.48 (0.4, 0.56)	0.18 (0.03, 0.33)	0.3 (0.21, 0.39)	0.38 (0.29, 0.46)	5.25 (3,193)	**0.002**
**Serum P1NP** **(ng/ml)**	156.32 (134.19,178.45)	39.35 (−2.43, 81.12)	76.86 (52.17, 101.55)	65.18 (42.89, 87.48)	15.33 (3,194)	**<0.001**

Data described as adjusted mean (95% CI).

**Table 10 diagnostics-10-01075-t010:** Pairwise comparison of mean bone turnover markers in between groups (ANOVA). Multiple comparisons using Games–Howell method.

Group Comparison	Adjusted Mean Difference (95% CI)	*p*-Value
**Serum CTX**		
Hyperthyroid vs. Hypothyroid	0.3 (0.16, 0.44)	**<0.001**
Hyperthyroid vs. Euthyroid	0.2 (0, 0.4)	0.054
Hyperthyroid vs. Control	0.1 (−0.04, 0.23)	0.236
Hypothyroid vs. Euthyroid	−0.1 (−0.28, 0.08)	0.481
Hypothyroid vs. Control	−0.2 (−0.3, −0.1)	**<0.001**
Euthyroid vs. Control	−0.1 (−0.28, 0.07)	0.426
**Serum P1NP**		
Hyperthyroid vs. Hypothyroid	115.48 (76.04, 154.92)	**<0.001**
Hyperthyroid vs. Euthyroid	83.37 (27.95, 138.78)	0.001
Hyperthyroid vs. Control	89.01 (49.75, 128.26)	**<0.001**
Hypothyroid vs. Euthyroid	−32.12 (−75.35, 11.11)	0.213
Hypothyroid vs. Control	−26.48 (−43.16, −9.8)	**0.001**
Euthyroid vs. Control	5.64 (−37.43, 48.71)	0.986

Data described as adjusted mean (95% CI).

**Table 11 diagnostics-10-01075-t011:** Post hoc pairwise comparison between mean of bone turnover markers adjusted for age and gender in between groups (ANCOVA).

Group Comparison	Adjusted Mean Difference (95% CI)	*p*-Value
**Serum CTX**		
Hyperthyroid vs. Hypothyroid	0.31 (0.07, 0.54)	**0.004**
Hyperthyroid vs. Euthyroid	0.18 (0.01, 0.34)	0.028
Hyperthyroid vs. Control	0.12 (−0.04, 0.28)	0.280
Hypothyroid vs. Euthyroid	−0.13 (−0.37, 0.11)	0.926
Hypothyroid vs. Control	−0.19 (−0.42, 0.05)	0.205
Euthyroid vs. Control	−0.06 (−0.23, 0.11)	>0.95
**Serum P1NP**		
Hyperthyroid vs. Hypothyroid	117.02 (52.97, 181.08)	**<0.001**
Hyperthyroid vs. Euthyroid	79.38 (34.35, 124.41)	**<0.001**
Hyperthyroid vs. Control	91.38 (48.29, 134.48)	**<0.001**
Hypothyroid vs. Euthyroid	−37.64 (−103.73, 28.45)	0.784
Hypothyroid vs. Control	−25.64 (−89.9, 38.62)	>0.95
Euthyroid vs. Control	12 (−34.47, 58.47)	>0.95

Data described as adjusted mean (95% CI).

**Table 12 diagnostics-10-01075-t012:** ANOVA analysis. Comparison the mean of Z score for Hip and Spine, and bone density (g/cm^2^) for Hip and Spine in between groups.

Variables (*n*)	Hyperthyroid (57)	Hypothyroid (15)	Euthyroid (50)	Control (64)	F-Stat (df)	*p* Value
Hip Z score	−0.02 (0.83)	0.25 (0.95)	0.27 (1.15)	0.39 (1.03)	1.72 (3,182)	0.164
Spine Z score	−0.29 (0.93)	−0.65 (0.79)	−0.18 (1.09)	−0.48 (1.03)	1.3 (3,182)	0.275
Hip g/cm^2^	0.86 (0.11)	0.93 (0.14)	0.91 (0.15)	0.95 (0.13)	5.12 (3,182)	**0.002**
Spine g/cm^2^	0.97 (0.09)	0.97 (0.09)	1.01 (0.13)	1.00 (0.13)	1.197 (3,182)	0.312

Data described as mean (SD).

**Table 13 diagnostics-10-01075-t013:** Pairwise comparison of Z score and bone density (g/cm^2^) of hip and spine among groups (ANOVA). Multiple comparisons using Bonferroni method.

Group Comparison	Adjusted Mean Difference (95% CI)	*p*-Value
**Hip (Z Score)**		
Hyperthyroid vs. Hypothyroid	−0.27 (−1.05, 0.51)	>0.95
Hyperthyroid vs. Euthyroid	−0.29 (−0.81, 0.23)	0.845
Hyperthyroid vs. Control	−0.41 (−0.9, 0.08)	0.162
Hypothyroid vs. Euthyroid	−0.02 (−0.81, 0.77)	>0.95
Hypothyroid vs. Control	−0.14 (−0.91, 0.63)	>0.95
Euthyroid vs. Control	−0.12 (−0.63, 0.39)	>0.95
**Spine (Z Score)**		
Hyperthyroid vs. Hypothyroid	0.36 (−0.42, 1.13)	>0.95
Hyperthyroid vs. Euthyroid	−0.11 (−0.62, 0.41)	>0.95
Hyperthyroid vs. Control	0.19 (−0.3, 0.67)	>0.95
Hypothyroid vs. Euthyroid	−0.46 (−1.25, 0.32)	0.713
Hypothyroid vs. Control	−0.17 (−0.94, 0.6)	>0.95
Euthyroid vs. Control	0.29 (−0.21, 0.8)	0.743
**Hip (g/cm^2^)**		
Hyperthyroid vs. Hypothyroid	0.08 (−0.18, 0.03)	0.248
Hyperthyroid vs. Euthyroid	−0.05 (−0.12, 0.01)	0.174
Hyperthyroid vs. Control	−0.09 (−0.15, −0.03)	**<0.001**
Hypothyroid vs. Euthyroid	0.02 (−0.09, 0.14)	0.944
Hypothyroid vs. Control	−0.02 (−0.13, 0.10)	0.974
Euthyroid vs. Control	− 0.04 (−0.11, 0.03)	0.445
**Spine (g/cm^2^)**		
Hyperthyroid vs. Hypothyroid	−0.01 (−0.07, 0.07)	>0.95
Hyperthyroid vs. Euthyroid	−0.04 (−0.10, 0.02)	0.337
Hyperthyroid vs. Control	−0.03 (−0.08,0.02)	0.463
Hypothyroid vs. Euthyroid	−0.04 (−0.012, 0.04)	0.607
Hypothyroid vs. Control	−0.03 ( −0.11, 0.55)	0.731
Euthyroid vs. Control	0.01 (−0.06, 0.07)	0.992

Data described as adjusted mean (95% CI).

**Table 14 diagnostics-10-01075-t014:** ANCOVA analysis adjusts for age and gender. Comparisons of the Z score mean for hip and spine and bone density (g/cm^2^) for hip and spine in between groups.

Variables *(n)*	Hyperthyroid (57)	Hypothyroid (15)	Euthyroid (50)	Control (64)	F-Stat (df)	*p* Value
Hip Z score	0.02 (−0.26, 0.3)	0.28 (−0.24, 0.79)	0.24 ( −0.06, 0.55)	0.43 (0.18, 0.68)	1.62 (3,180)	0.187
Spine Z score	−0.26 (−0.54, 0.02)	−0.62 (−1.14, −0.11)	−0.17 (−0.47, 0.13)	−0.46 (−0.71, −0.21)	1.24 (3,180)	0.588
Hip g/cm^2^	0.877 (0.84, 0.92)	0.95 (0.88, 1.02)	0.91 (0.88, 0.96)	0.96 (0.93, 0.99)	3.81 (3,177)	**0.011**
Spine g/cm^2^	0.98 (0.94, 1.02)	0.98 (0.91, 1.05)	1.01 (0.97, 1.05)	1.01 (0.98, 1.04)	0.60 (3,177)	0.617

Data described as adjusted Mean (95% CI).

**Table 15 diagnostics-10-01075-t015:** Pairwise comparison of Z score and bone density (g/cm^2^) of hip and spine among groups adjust for age and gender (ANCOVA).

Group Comparison	Adjusted Mean Difference (95% CI)	*p*-Value
**Hip (Z Score)**		
Hyperthyroid vs. Hypothyroid	−0.26 (−1.03, 0.51)	>0.95
Hyperthyroid vs. Euthyroid	−0.23 (−0.75, 0.30)	0.845
Hyperthyroid vs. Control	−0.41 (−0.90, 0.09)	0.178
Hypothyroid vs. Euthyroid	−0.03 (−0.76, 0.82)	>0.95
Hypothyroid vs. Control	−0.15 (−0.92, 0.62)	>0.95
Euthyroid vs. Control	−0.18 (−0.71, 0.35)	>0.95
**Spine (Z Score)**		
Hyperthyroid vs. Hypothyroid	0.36 (−0.42, 1.14)	>0.95
Hyperthyroid vs. Euthyroid	−0.09 (−0.62, 0.44)	>0.95
Hyperthyroid vs. Control	0.20 (−0.30, 0.70)	>0.95
Hypothyroid vs. Euthyroid	−0.45 (−1.25, 0.34)	0.777
Hypothyroid vs. Control	−0.16 (−0.94, 0.61)	>0.95
Euthyroid vs. Control	0.29 (−0.24, 0.82)	0.849
**Hip (g/cm^2^)**		
Hyperthyroid vs. Hypothyroid	−0.07 (−0.15, 0.01)	0.521
Hyperthyroid vs. Euthyroid	−0.04 (−0.12, 0.04)	>0.95
Hyperthyroid vs. Control	−0.09 (−0.14, −0.03)	**<0.001**
Hypothyroid vs. Euthyroid	0.04 (−0.88, 0.15)	>0.95
Hypothyroid vs. Control	−0.01 (−0.12, 0.10)	>0.95
Euthyroid vs. Control	−0.05 (−0.12, 0.02)	0.481
**Spine (g/cm^2^)**		
Hyperthyroid vs. Hypothyroid	−0.01( −0.11, 1.00)	>0.95
Hyperthyroid vs. Euthyroid	−0.03( −0.10, 0.05)	>0.95
Hyperthyroid vs. Control	−0.03 (−0.10,0.04)	>0.95
Hypothyroid vs. Euthyroid	−0.02 (−0.13, 0.08)	>0.95
Hypothyroid vs. Control	−0.02 ( −0.12, 0.08)	>0.95
Euthyroid vs. Control	0.01 (−0.07, 0.07)	>0.95

Data described as adjusted mean (95% CI).

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
