# Peer review of "Comparison of Vitamin D Levels, Bone Metabolic Marker Levels, and Bone Mineral Density among Patients with Thyroid Disease: A Cross-Sectional Study"

_diagnostics, 2020, doi:10.3390/diagnostics10121075_

Round 1

Reviewer 1 Report

The study is well designed and carried out. The article is well written. The results are clear and add substantially to our knowledge of the thyroid status and bone metabolism.

Author Response

Dear Sir,

Thank you for giving us the opportunity to submit a revised draft of the manuscript “Comparison of Vitamin D Levels, Bone Metabolic Marker Levels, and Bone
Mineral Density among Patients with Thyroid Disease: A Cross-Sectional Study
(Manuscript ID: diagnostics-957293).

We appreciate the time and effort that you dedicated to providing feedback on our manuscript and are grateful for the insightful comments.

We look forward to hearing from you regarding our submission.

Thank You.

Reviewer 2 Report

The relation between thyroid disease and bone health is well described, in particular  the effects of hyperthyroidism on bone density.  However, the effects of 25OHD on this relation or the effect of 25OHD insufficiency on the incidence of thyroid disease is less clear; this study extends to these findings to a predominantly Malay population.

The cross-sectional observational study reports essentially negative findings – with  evidence for some of the described effects of hyperthyroidism on bone density, but no evidence for a role of 25OHD insufficiency on the incidence of thyroid disease or the coincidence of bone disease.

I have one major issue –

It is not universally accepted that a serum vitamin D concentration of < 50 nmol/L is “low” – there is better evidence of 30 nmol/L.  This classification is controversial and may not lead to health benefits.  The authors need to justify the use of this cut-off or re-assess their conclusions accordingly.

Minor issues

The authors use mean and SD to describe measured variables which they state are gaussian - I cannot see age being normally distributed, and TSH is also rarely gaussian. 

The authors frequently use the the term ‘normal’ to describe parameters within the reference interval – eg The mean serum calcium level was normal in all groups.  I think “The mean serum calcium level was within the 95% reference interval normal in all groups” is preferable.

The authors quote reference 5 "In India, hyperthyroid patients with concomitant vitamin D deficiency had a lower BMD than those
60 with adequate vitamin D levels" -whilst I appreciate this is a rather smalll cohort is such sub group analysis possible, as I think the key question is does 25OHD insufficiency pre-dispose to thyroid disease associated bone loss

Scatter plots may be enlightening for some of the significant differences.

Author Response

Dear Sir,

Thank you for giving us the opportunity to submit a revised draft of the manuscript “Comparison of Vitamin D Levels, Bone Metabolic Marker Levels, and Bone Mineral Density among Patients with Thyroid Disease: A Cross-Sectional Study(Manuscript ID: diagnostics-957293).

We appreciate the time and effort that you dedicated to providing feedback on our manuscript and are grateful for the insightful comments.

We have incorporated most of the suggestions made and those changes are highlighted in "yellow" within the manuscript. Please in the attachment file, the response to your comments and concerns. All page numbers refer to the revised manuscript file with tracked changes.

We look forward to hearing from you regarding our submission.

Thank You.

Kind regards

Dr. Tuan Salwani Tuan Ismail

Reviewer 3 Report

General

 Thank you very much for the opportunity to review this manuscript. I read it with great interest. However, almost all references are listed without volume and page numbers. The authors must correct them first and foremost.

Specific

Abstract

  1. Line 37-38 – This was already described in line 32-33. I recommend deleting it.
  2. Line 40-41 – This was already described in line 36-37. I suggest modifying it to be the conclusion of this study.

Introduction

  1. Line 50-52 – Is it true “thyroid hormones”, not “para-thyroid hormones”?

Materials and Methods

  1. Line 82-83 – Please provide the diagnostic criteria of euthyroid patients.
  2. Line 113-114 – The authors determined three classifications of vitamin D levels; however, only the number of vitamin D insufficient (n=145 [73%]) and the whole range 12.26 to 111.22 nmol/L were provided. Please provide more summarized details in Table 1 or the other appropriate place.
  3. Line 115 – Please change “CV” to “coefficient of variance (CV)”.

Discussion

  1. Line 335-350 – The authors explained two possibilities of insufficient serum vitamin D levels through all participants. One of them was insufficient consuming vitamin D-fortified food or vitamin D supplements. Another was insufficient sunlight exposure. As the authors mentioned the latter possible cause, the characteristics of the participants were predominantly important to discuss the present results. Please provide the characteristics of the participants, such as how many hospital staff was included in the control subjects, and so on.
  2. Line 351-358 – The authors mentioned two possible confounders, such as living area and seasonal change. To compare the previous studies, please provide living area information and a sub-analysis of vitamin D level, in which the participants were stratified by recruited season.
  3. Line 386-387 – The authors mentioned the differences in the patient selection process might cause the discrepancies in the relationship of the 25(OH)D levels between this study the previous ones. I agreed with you because the male serum 25(OH)D level in the control group (44.82 nmol/L) was extremely lower than the others. Male participants were 42.2% of the control groups, which was 1.7-2.0 times higher than the other groups. Please discuss the reason why the serum 25(OH)D level in the male control group was so lower than the others.

Author Response

Dear Sir,

Thank you for giving us the opportunity to submit a revised draft of the manuscript “Comparison of Vitamin D Levels, Bone Metabolic Marker Levels, and Bone Mineral Density among Patients with Thyroid Disease: A Cross-Sectional Study(Manuscript ID: diagnostics-957293).

We appreciate the time and effort that you and the reviewers dedicated to providing feedback on our manuscript and are grateful for the insightful comments on and valuable improvements to our paper. We have incorporated most of the suggestions made and those changes are highlighted in "blue" within the manuscript. Please in the attachment file, the response to your comments and concerns. All page numbers refer to the revised manuscript file with tracked changes.

We hope that you will find our amended manuscript sufficiently improved for publication. We look forward to hearing from you regarding our submission and to respond to any further questions and comments you may have.

Thank you

Kind regards

Dr. Tuan Salwani Tuan Ismail

Round 2

Reviewer 3 Report

General

 Thank you for your response; however, almost all references are still listed without volume and page numbers. Only references 1, 16, 20, 23, 30, 37, and 45 seem to be all right. I suggest that the authors must correct them first and resubmit again.

Specific

Abstract

  1. Line 44-45 – The sentence in line 40-41 of the previous manuscript was deleted; however, no conclusion of this study was found. The abstract should contain a conclusion, as described in the instruction for authors.

Introduction

  1. Line 54-56 – The sentence seemed to be cited the reference 2. The sentence cited by the authors was described based on reference 5, “Mosekilde L, Christensen MS. Decreased parathyroid function in hyperthyroidism: inter relationship between serum parathyroid hormone, calcium phosphorus metabolism and thyroid function. Acta Endocrinol (Cophenh) 1977;84:566-575.” In the original literature of Mosekild and Christensen, no thyroid hormone was assessed, and I cannot find any description of the function of thyroid hormone for leading to increased bone mineral resorption and calcium loss via the kidneys. I suggest the sentence should be deleted. Additionally, the reference had no volume and page number (15(Suppl 2): S107-112) in the author’s response.

Materials and Methods

  1. Line 89-90 – The authors should provide the adult’s reference interval values.

Results

  1. Line 164-169 and Table 1 – The values of “Duration of thyroid disease”, “TSH”, and “FT4” were inconsistent.

Discussion

  1. Line 357-360 – The characteristics of the healthy control group should be described in the Materials and Methods section. In there, the effect of the difference in sunlight exposure among the subjects must be discussed with the analysis results described in the Results section. If you didn’t have enough information about the sunlight exposure of the subjects, I recommend mentioning as one of the limitations of this study.
  2. Line 374-377 – The authors should provide a sub-analysis of vitamin D level, in which the participants were stratified by recruited season. If you have no information about the date of the sample collection, I recommend mentioning it as one of the limitations of this study.
  3. Author’s response to my comment 11, about the previous manuscript line 374-377 – The authors mentioned “we already highlighted this issue. (338-341 and 355-358).” Meanwhile, my comment was “please discuss the reason why the serum 25(OH)D level in the male control group was so lower than the others.” The former may the description of consuming vitamin-D-fortified food or vitamin D supplements. However, any information about the difference between the male control group and the others in vitamin D intake was provided. I don’t think the authors already highlighted this issue. Meanwhile, the latter may the description of sunlight. The authors provided new information that 75% of hospital staff who had less exposure to sunlight in the control group. If the including hospital staffs were the cause of the lower serum 25(OH)D level, the selection of the control groups might be inappropriate. I suggest the authors must discuss the validity of the control subject selection.

Author Response

Dear reviewer,

On behalf of my co-authors, I would like to thank you for the opportunity to revise and resubmit our manuscript “Comparison of Vitamin D Levels, Bone Metabolic Marker Levels, and Bone Mineral Density among Patients with Thyroid Disease: A Cross-Sectional Study” (Manuscript ID: diagnostics-957293).

We found your comments to be helpful in revising the manuscript and we have carefully considered and responded to each suggestion.  We have included a response to reviewers in which we address each comment the reviewers made as in attachment.

Those changes are highlighted in “grey” in the manuscript. All page numbers refer to the revised manuscript file with tracked changes. In this current revised manuscript, we also rearrange the paragraphs in the discussion part to make them more orderly.

Thank you again for your consideration of our revised manuscript. We hope that you will find our amended manuscript sufficiently improved for publication.

Sincerely,

Dr Tuan Salwani Tuan Ismail

Department of Chemical Pathology

School of Medical Sciences

Universiti Sains Malaysia Health Campus

16150 Kota Bharu

Kelantan, Malaysia

Phone: +6013 6647129/+6097676510

Round 3

Reviewer 3 Report

The paper has been improved in accordance with the comments.

I suggest the conclusion in the Abstract should be modified more sophisticated one, like the Conclusion section in the manuscript.

Author Response

Dear reviewer,

Thank you for pointing this out. The reviewer is correct, and we have made a modification. The revised text reads as follows on line 46-47: “In conclusion, this study showed no correlation between serum 25(OH)D levels and thyroid diseases. The bone markers showed a difference between thyroid groups with no significant difference in BMD.”

Thank you again for your consideration of our revised manuscript. We hope that you will find our amended manuscript sufficiently improved for publication.

Sincerely,

Dr Tuan Salwani Tuan Ismail

Department of Chemical Pathology

School of Medical Sciences

Universiti Sains Malaysia Health Campus

16150 Kota Bharu

Kelantan, Malaysia

Phone: +6013 6647129/+6097676510